# Bioelectrical Impedance Analysis Versus Dual X-Ray Absorptiometry for Obesity Assessment in Pediatric Populations: A Systematic Review

**DOI:** 10.3390/diagnostics15121505

**Published:** 2025-06-13

**Authors:** Lorena Mihaela Manole, Gabriela Ghiga, Otilia Iftinchi, Laura Otilia Boca, Mădălina Andreea Donos, Elena Țarcă, Nistor Ionuț, Ninel Revenco, Iulia Margasoiu, Laura Mihaela Trandafir

**Affiliations:** 1Department of Mother and Child, Faculty of Medicine, Grigore T. Popa University of Medicine and Pharmacy, 700115 Iasi, Romania; lorena.manole@umfiasi.ro (L.M.M.); ghiga.gabriela@yahoo.com (G.G.); otilia.iftinchi@yahoo.com (O.I.); laura.boca@umfiasi.ro (L.O.B.); mada.donos@yahoo.com (M.A.D.); iuliamargasoiu@yahoo.ro (I.M.); laura.trandafir@umfiasi.ro (L.M.T.); 2Saint Mary Emergency Children Hospital, 700309 Iasi, Romania; 3Department of Pediatric Surgery and Orthopedics, Faculty of Medicine, “Grigore T. Popa” University of Medicine and Pharmacy, 700115 Iasi, Romania; 4Department of Medicine, Faculty of Medicine, Grigore T. Popa University of Medicine and Pharmacy, 700115 Iași, Romania; ionut.nistor@umfiasi.ro; 5Department of Nephrology, “Dr. C.I. Parhon” Clinical Hospital, 700503 Iasi, Romania; 6Department of Pediatrics, University of Medicine and Pharmaceutics “Nicolae Testemițanu”, MD-2001 Chișinău, Moldova; ninel.revenco@usmf.md

**Keywords:** childhood obesity, body composition, bioelectrical impedance, dual X-ray absorptiometry, pediatric

## Abstract

**Objectives**: Pediatric obesity represents a significant public health challenge, requiring accurate and accessible tools for assessing body composition in pediatric populations. This systematic review (PROSPERO CRD42024592366) compares the methodological accuracy and clinical utility of bioelectrical impedance analysis (BIA) and dual x-ray absorptiometry (DXA) in evaluating obesity among children and adolescents. **Methods**: Utilizing a comprehensive search across PubMed, EMBASE, and Web of Science between 1 January 2014 and 31 December 2024, we identified 28 studies meeting our inclusion criteria. The studies included involved participants aged 2–17 years with obesity and compared BIA with DXA as the reference standard. The exclusion criteria were studies focusing on adults, those that assessed BC solely using anthropometry, and those that did not report primary outcomes relevant to the comparison of BIA vs. DXA. **Results**: The findings reveal that despite recent technological advances improving BIA’s precision, it consistently underestimates body fat percentage and fat mass, particularly in overweight and obese pediatric populations. DXA it is often used as a reference method in the evaluation of whole-body composition due to its higher accuracy and reliability. BIA offers significant practical advantages in accessibility, cost-effectiveness, and portability, but enhancements are needed to improve its accuracy for individual-level assessments. **Conclusions**: While BIA shows promise as a practical tool for body composition assessment in children, its accuracy varies significantly by device type. Multi-frequency segmental analyzers, such as InBody 720, demonstrate better agreement with DXA, whereas simpler models tend to underestimate fat mass. Therefore, conclusions regarding BIA performance should be device-specific and its clinical utility should be carefully weighed based on the technology used.

## 1. Introduction

Obesity is a complex and multifactorial condition that is increasingly prevalent worldwide, affecting both pediatric and adult populations. Its rising prevalence poses significant risks of comorbidity and carries substantial social and economic consequences. This disease in children can significantly affect their entire well-being, increasing the risk of major health issues such as cardio-metabolic diseases, as well as their psychological and overall physical health [1,2,3,4]. Body assessment methods at the moment are numerous and have been discussed in depth. Traditional metrics, such as body mass index (BMI), have limitations in distinguishing between fat mass (FM) and fat-free mass (FFM) [5], leading to an increased need for more precise body composition (BC) assessment tools.

Dual-energy X-ray absorptiometry (DXA) is considered the gold standard for the determination of bone density, but it is often used as a reference method in the evaluation of whole-body composition, because it provides accurate measures of body FM and lean mass [1,6,7]. However, this method is expensive, difficult to implement on a large scale [1,8], requires specialized training, and involves radiation exposure, making it less practical for routine clinical use and repeated assessments in pediatric populations. Additionally, there is a risk of irradiation if the patient undergoes repeated scans over time. When discussing the process of weight loss guided by health professionals, we also emphasize the need to reevaluate these children and their parameters at least once a month. Current pediatric guidelines recommend that children with obesity be followed by clinicians as early as possible until adulthood to prevent the occurrence of pathologies such as type 2 diabetes, cardiovascular diseases, and metabolic-dysfunction-associated steatotic liver disease [1].

Bioelectrical impedance analysis (BIA) has gained popularity as an accessible, non-invasive, low-cost alternative capable of assessing body composition efficiently in clinical and epidemiological settings [9,10,11,12]. Despite technological advances, the accuracy of BIA compared to DXA remains uncertain [13], particularly for pediatric obesity, in which variations in hydration and tissue composition can significantly impact accuracy [14,15]. There is a great variety of BIA devices which can be classified according to the frequency, electrode placement, device type, and output mode. By frequency, the types are as follows: single-frequency BIA (SF-BIA), multi-frequency BIA (MF-BIA), and bioelectrical impedance spectroscopy (BIS). The accuracy of older BIA models using a single frequency can be compromised by factors such as hydration status and food intake, because dehydration can slow the current’s movement through the body [5]. BIA devices that use multiple-frequency electrical currents can more accurately differentiate between intracellular and extracellular water compared to single-frequency devices. This improved differentiation enhances the accuracy of BC measurements [16].

In both methods, certain components are measured directly, such as impedance in BIA or X-ray attenuation in DXA, while others, like percent body fat (BF%), FFM, or skeletal muscle mass, are estimated using device-specific predictive algorithms. BIA can be useful for field studies and clinical screenings in children and adolescents because it is portable, affordable, and fast. Its accuracy, however, is affected by factors like age, genre, hydration status, body position, and the type of BIA technology used [5]. DXA offers greater accuracy at the individual level but is less practical for repeated or large-scale use in pediatric settings. In the current scientific literature, many articles on BIA and DXA present contradictory information regarding the evaluation of BC among pediatric populations. This systematic review aims to critically evaluate existing evidence comparing the methodological accuracy and clinical utility of BIA and DXA, providing health professionals and researchers with clear guidance for pediatric obesity assessment.

## 2. Materials and Methods

### 2.1. Protocol Registration and Reporting

This systematic review followed PRISMA (Preferred Reporting Items for Systematic Review and Meta-Analysis) guidelines [17]. In order to realize this systematic review, the authors formulated a search protocol regarding the assessment of BC registered at PROSPERO (CRD42024592366).

### 2.2. Search Strategy and Selection Process

The literature search was performed in PubMed, EMBASE, and Web of Science from 1 January 2014 to 31 December 2024, using the following keywords: ‘bioelectrical impedance’ AND ‘dual X-ray absorptiometry’ AND ‘obesity’ AND ‘child’ OR ‘children’ OR ‘adolescent*’ OR ‘pediatric’ OR ‘childhood’. The specific search strategies were tailored to the requirements of each database. Only English-language studies involving human pediatric populations were included. We used the PRISMA checklist and flowchart to ensure a high-quality search and to minimize bias. For the duplicate data set studies in all the databases, only one study from each set was included. Additionally, a cross-referencing search was performed for the full-text versions of the preliminarily included articles. Additionally, the references of the identified studies were manually searched to ensure that no study was missed.

All decisions regarding the inclusion or exclusion of papers were made according to a consensus, based on the predefined criteria by two authors. The inclusion criteria for the study were as follows: studies assessing BC in pediatric subjects aged between 2 and 17 years with obesity; studies directly comparing BIA and DXA; cross-sectional, longitudinal, observational, and randomized controlled trials; studies with full-text availability in English. Studies comparing interventions in pediatric patients with obesity to those without obesity were included. Exclusion criteria were as follows: studies focusing on adults; studies assessing BC solely using anthropometry; and studies that do not report primary outcomes relevant to the comparison of BIA vs. DXA.

### 2.3. Data Extraction

To achieve the research objectives, we conducted a systematic literature review and assessed intervention effects using the PICO (Population, Intervention, Comparison, Outcome) framework. The Population consisted of pediatric subjects aged 2–17 years with obesity. The Intervention involved DXA, recognized as the “gold standard” for BC assessment, alongside various BIA devices. The Comparison focused on evaluating obesity and BC assessment outcomes obtained through these two methods. The Outcome aimed to validate the accuracy of BIA devices and assess their clinical utility in pediatric obesity research.

Two independent researchers systematically reviewed the titles, abstracts, and full texts of relevant studies. Articles that met the predefined inclusion criteria were independently evaluated to ensure methodological rigor and data reliability.

Extracted data were systematically organized in a Microsoft Excel spreadsheet using a predefined checklist. Key variables included the following: author name, publication year, country of study, study design, sample size, intervention type, follow-up duration, assessment techniques (devices used), comparison details, methodological accuracy, primary outcomes, and clinical utility. The studies included in this review utilized a variety of BIA devices compared to DXA. Most BIA devices used manufacturer-specific predictive algorithms to estimate FM, FFM, and total body water. Also, they did not disclose the exact equations used; however, some studies noted the use of formulas developed and validated for pediatric populations by the manufacturers. The DXA devices analyzed BC using the manufacturers’ proprietary software, which applies internal algorithms to distinguish between bone mineral content, lean tissue, and FM. The studies included typically did not report customized or external estimation equations, relying on standard software outputs.

Due to variations in study design, participant characteristics, and measurement tools, a meta-analysis was not feasible. Instead, a comprehensive narrative synthesis was conducted to evaluate the validity of BIA compared to DXA for BC assessment in obese children and adolescents. Studies provided data on pediatric subjects, including age- and sex-based subgroups, as well as obesity assessments using BIA, DXA, or a combination of both methods.

In each study, data were analyzed by subgroup and assessment method to determine measurement precision and accuracy. In some cases, weight categories such as underweight, normal weight, overweight, and obesity were combined due to limited data, preventing meaningful stratification. All extracted data were synthesized manually to ensure a thorough and structured presentation of findings based on participant groups and study characteristics.

### 2.4. Quality Assessment

Two authors independently evaluated the risk of bias using the Quality Assessment of Diagnostic Accuracy Studies (QUADAS−2) tool [18]. This method is widely recommended for systematic reviews to assess the risk of bias and applicability of primary diagnostic accuracy studies. This tool assesses four domains: patient selection, index test, reference standard, and flow and timing. Each domain is rated based on a predefined set of criteria that consider methodological aspects. Consensus was reached between the reviewers for all assessments, with no need for a third reviewer to resolve discrepancies.

## 3. Results

A total of 544 studies were initially identified through the literature search of studies published between 2014 and 2024. After scanning the titles, excluding non-relevant studies and duplicates, and conducting a full-text review, 28 articles met the inclusion criteria. The PRISMA flowchart (Figure 1) details the selection process. Table 1, Table 2, Table 3, Table 4 and Table 5 provide a structured summary of the included studies, grouped according to the specific BIA device manufacturer (Tanita, InBody, SECA, Bodystat, and Quantum). Each table includes study details such as country, sample size, intervention type (if any), the BIA and DXA devices used, key findings, and clinical applicability. This device-based organization system highlights the methodological variability and diagnostic performance differences across technologies, which is essential for interpreting the validity and clinical utility of BIA compared to DXA in pediatric populations.

Various types of articles are included in this review. The type of study design and the statistical significance analysis of the included articles can be seen in Table 6.

Across the studies, BIA consistently underestimated FM and %BF compared to DXA, notably among overweight and obese individuals, while overestimating the FFM. The mean bias in FM ranged from −2.6 to −9.9%, with the correlation coefficients between these methods being generally high (r = 0.61–0.99), though clinically relevant biases remained.

Clinical utility varied significantly by device. Tanita MC-180MA and SECA devices provided adequate accuracy at a group level but showed limitations for individual assessments. Devices like Tanita BC-418, BC-532, InBody 720, and Quantum III demonstrated significant accuracy issues, particularly with increasing adiposity. Octopolar, multi-frequency devices like InBody 370 were comparatively more precise, particularly in severely obese children, providing clinically acceptable estimates for appendicular lean mass and total body FM. The SKF thicknesses and 4C model equations were determined by researchers or special technicians, and the other methods used built-in machine measurements.

The risk of bias was assessed using the Quality Assessment of Diagnostic Accuracy Studies (QUADAS-2) tool [18]. This tool assessed the four domains for each study (Figure 2): patient selection, index test, reference standard, and flow and timing.

BIA showed strong correlations with DXA in tracking changes over time, making it useful for longitudinal studies but less reliable for precise individual assessments. Multi-frequency BIA devices demonstrated an improved accuracy over single-frequency models, though hydration status remained a confounding factor.

In the clinical utility of methods, the authors draw attention to the accuracy of the BIA method. The Tanita TBF-310 [21], Tanita BC-418 [22,23,24,26,27,28], Tanita BC-532 [32], InBody 720 [37,38,39], InBody 770-BIA [5], and BIA Quantum [44] tetrapolar bioelectrical impedance analyser and Quantum III [45] devices are considered inaccurate or not valid for clinical or epidemiological use and must be applied with caution in the assessment of children and adolescents with obesity. The Tanita MC-180MA [19] is considered to be a valuable clinical tool to measure BC at the group level, but inaccurate for individual obese adolescents. The Tanita DC-430S MA device [30], BIA SECA mBCA514 (without the handrail method because it is a less accurate method) and SECA-525 [40], the Omron handheld single-frequency tetrapolar device, and InBody 370 [36] are described to be precise in estimating BF% and appendicular lean mass compared to DXA, which makes them very useful in clinics for evaluating BC in children with obesity.

Despite these BIA limitations, some authors consider Tanita SC-240A [20], Tanita BF-689 [9,33], and Bodystat Quadscan 4000 [42,43] as viable alternatives to DXA for measuring BC in obese children. When interpreting individual results, the validity of these methods may vary depending on sex and weight status. However, they can still be regarded as viable alternatives for monitoring changes in adiposity over time and have high specificity for classifying individuals of normal weight and those with obesity.

## 4. Discussion

In both clinical practice and research, there is a growing demand for practical, precise, and cost-effective tools to assess BC in overweight and obese children and adolescents. Given the high prevalence of childhood obesity worldwide, integrating nutritional medical guidance into screening programs could enhance early detection and preventive strategies [46].

Research indicates that both the BMI and DXA methods have notable drawbacks. DXA remains superior in accuracy, particularly for individual assessments, and is critical for clinical intervention and follow-up. However, due to its cost, limited accessibility, and radiation concerns [47], its routine or repeated use remains impractical in clinical and community settings. Conversely, BIA’s ease-of-use, affordability, and portability make it attractive, especially for large-scale or epidemiological purposes. It has the potential to serve as a valuable clinical tool for group-level assessments [19]. Yet, significant inaccuracies persist, especially among obese children, necessitating a cautious interpretation of BIA results in individual clinical contexts.

Establishing the accuracy of various BIA devices in obese individuals is crucial, as increased adiposity affects tissue hydration and FM determination. In the literature, there is a high correlation between BIA and DXA methods, but the precise estimates significantly vary across devices, BMI categories [37], sex and age groups, groups by hydration status and food intake [5], physical activity levels [31], and the sensitivity and specificity of the methods of assessment.

Kabiri et al. [9] and Butcher et al. [33] both evaluated the Tanita BF-689 device. While Kabiri et al. highlighted its reliability and classification accuracy for overweight/obese children, Butcher et al. specifically confirmed its effectiveness in tracking %BF changes in adolescents aged 12–17. The Tanita BF-689 device demonstrated a high accuracy in classifying adolescents based on %BF, correctly categorizing 79% of participants as healthy, making it a useful tool for both healthcare professionals and parents in monitoring the BC of children and adolescents.

In the majority of studies, DXA is considered the reference method for validating BIA measurements, because it is more sensitive to BC changes with built-in machine measurements [23]. BIA’s accuracy varies due to differences in built-in equations, which rely on impedance measurements that can be influenced by hydration levels, body fat distribution, and muscle mass. Seo et al. reported that as BMI and FM increase, muscle mass (which contains a large amount of water) decreases relatively, affecting the proportion of total body water. This, in turn, influences the accuracy of BIA because the built-in equations of the devices become less reliable [38].

Both the BIA and DXA methods provide valuable information about BC and BF%, which is essential for assessing obesity and monitoring changes in BC within special programs for losing weight and treatment [23,30,33]. They are even used as a screening tool for high-risk cardiometabolic comorbidities in the long term. However, it is crucial to consider the potential limitations and sources of error associated with each method, such as variations in hydration status affecting BIA measurements or radiation exposure with DXA [19,36]. Several studies indicate improvements in BIA accuracy through the development of population-specific predictive equations based on raw impedance data and anthropometric measures [48]. Future advancements should prioritize these customized equations and refine multi-frequency and segmental BIA technologies to improve clinical reliability [49].

A key limitation identified across the included studies is that most BIA devices use proprietary algorithms that are not publicly disclosed. As noted by Campa et al., these “black box” systems prevent users and researchers from knowing which prediction equations are used or whether they have been validated for specific populations, such as children and adolescents with obesity, metabolic syndrome, or dyslipidemia [50,51]. This lack of transparency restricts reproducibility and clinical relevance, particularly in pediatric populations that differ physiologically from adults. It also remains uncertain whether these devices account for pediatric and obesity data specifically in their algorithms, raising questions about how well their outputs apply to these groups. This methodological concern should be clearly acknowledged when comparing BIA and DXA results [36,51].

The limitations of our research include the high level of variability across study designs and the absence of standardized protocols for BIA. Differences in hydration, body composition equations, and device calibration also introduce measurement errors. Future studies should focus on standardizing protocols, developing and validating specific equations for different pediatric obesity subpopulations, and enhancing BIA technologies to improve accuracy and clinical relevance.

While BIA shows promise as a convenient and non-invasive method for obesity assessment, it may not yet be considered the new “standard method” compared to DXA. Although BIA is attractive due to its non-invasive, low-cost, and user-friendly nature, it remains an indirect method, as it estimates BC based on impedance, then interprets the result through assumptions built into its algorithms. Given this limitation, we do not propose BIA as a replacement for DXA. Instead, BIA may serve as a screening or monitoring tool in settings in which DXA is impractical, provided device-specific validation is performed and transparently reported. Healthcare providers should recognize BIA as a viable screening tool but reserve DXA for precise diagnostic evaluations, particularly in intervention studies or clinical settings demanding high accuracy. However, as BIA technology advances and validation studies continue to refine its accuracy, it may increasingly become a preferred method, particularly in settings in which access to DXA is limited. Further research and standardization efforts are needed to determine the role of BIA as a potential new standard method for obesity assessment.

## 5. Conclusions

Both bioelectrical impedance analysis and dual-energy x-ray absorptiometry have advanced significantly, with improved accuracy, speed, and clinical utility in assessing body composition in children and adolescents. However, their clinical applications differ based on factors such as precision, accessibility, and cost-effectiveness.

The findings of this systematic review suggest that while BIA can serve as a practical and non-invasive tool for estimating body composition in pediatric populations, its accuracy and agreement with DXA vary considerably depending on the specific device used. Therefore, BIA technology cannot be considered as a uniform method. Multi-frequency, segmental BIA devices (e.g., InBody 720 and SECA mBCA) demonstrated a higher level of concordance with DXA measurements compared to single-frequency or foot-to-foot analyzers, which showed greater variability and reduced precision.

Given this methodological heterogeneity, the clinical and research utility of BIA should be evaluated on a device-specific basis. Future research should avoid aggregating BIA data across devices and instead stratify analyses according to device type, frequency spectrum, and the inclusion of pediatric-specific prediction equations. Until cross-device standardization is established, BIA results should be interpreted with caution, guided by the technical characteristics of and validation evidence for the specific device used.

## Figures and Tables

**Figure 1 diagnostics-15-01505-f001:**
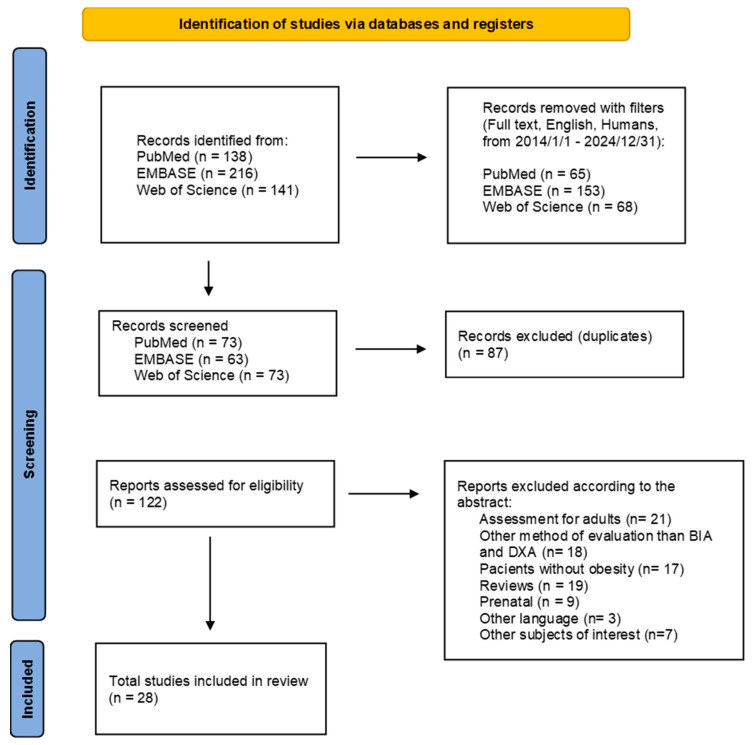
PRISMA flowchart, adapted after Page MJ at al. [17]. Abbreviations: BIA—bioelectrical impedance analysis; DXA—dual-energy X-ray absorptiometry.

**Figure 2 diagnostics-15-01505-f002:**
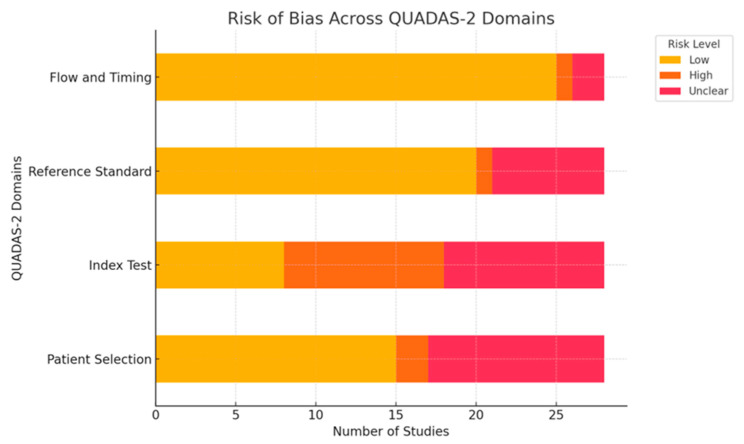
Risk of bias summary using QUADAS-2.

**Table 1 diagnostics-15-01505-t001:** BIA using Tanita devices.

Author, Year, Country	N	Intervention Type	Assessment Techniques (Devices)	Primary Outcome(s)	Clinical Utility
Wan C.S. et al., 2014 [19] Australia	66	Weight loss intervention	BIA (Tanita MC-180MA) (Tanita BIA8) DXA (GE-Lunar Prodigy)	The study aimed to compare BIA and DXA for assessing BC and tracking adiposity changes in overweight and obese adolescents. The results showed that BIA was less accurate for individual measurements but useful for group-level assessments.	The Tanita BIA8 device could be a valuable clinical tool to measure BC at the group level, but is inaccurate for individual obese adolescents.
de Silva M.H.A.D. et al., 2021 [20] Sri Lanka	97	No intervention	BIA (Tanita SC-240A) DXA scanner (Hologic Discovery W)	Significant mean differences were observed between DXA and BIA in measuring FM and FFM. Despite these differences, DXA and BMI-derived measurements for FM and FFM showed high correlations (FM r = 0.92 and FFM r = 0.83, both *p* < 0.001).	The errors of BIA accuracy were higher in boys compared to girls, indicating limitations of BIA in measuring BC. Despite these limitations, BIA remains a viable alternative to DXA for measuring BC in obese children aged 5–15 years. However, the accuracy errors should be considered when interpreting individual results.
Kasvis P. et al., 2014 [21] Canada	89	Family-centered lifestyle intervention	BIA (Tanita TBF-310) DXA (Hologic)	BIA accurately reflects the direction of changes in FM and FFM in overweight and obese children. However, the inaccuracy in the magnitude of BIA measurements may be attributed to differences in fat distribution patterns.	BIA can be used in a clinical setting to accurately measure direction of changes in FM and FFM over time, but cannot be used to accurately determine the magnitude of BC changes in overweight and obese children.
Gutierrez-Marín D. et al., 2021 [22] Spain	315	No intervention	BIA (Tanita BCe418MA) DXA (General Electric Lunar Prodigy Advance) ADP (BODPOD device) Four Compartment (4C) Equation	The predictive equation reduced the bias from the BIA outputs from 14.1% to 4.6%. The study found that BIA is a feasible tool for estimating BC, but the accuracy is subject to improvements via specialized equations.	The new predictive equation enhances the accuracy of BC assessment using BIA in obese children. The use of BIA, particularly with specific equations, facilitates BC assessment, without the need for expensive equipment or specialized training.
Martín-Matillas M. et al., 2020 [23] Spain	92	20-week exercise intervention	BIA (Tanita BC-418 MA), DXA (Hologic) SKF thicknesses for the Slaughter equations (Slg-Eq)	Girls experienced a greater underestimation than boys with the Slg-Eq method (*p* ≤ 0.001), and the extent of underestimation decreased the higher the participant’s weight status. Both BIA and Slg-Eq showed acceptable validity for tracking changes over time.	Both Slg-Eq and BIA are feasible for monitoring changes in adiposity, especially in large-scale or community settings, due to their ease of use. BIA’s accuracy improves with increased adiposity, but is less reliable in leaner children.
Vásquez F. et al., 2016 [24] Chile	61	No intervention	BIA (Tanita BC 418MA) DXA (Lunar Prodigy Ghc DPX-NT) Isotope dilution (Mass spectrometry) Plethysmography Four Compartment (4C) Equation [25]	The study aimed to evaluate the accuracy of body fat percentage (%BF) estimates across various BC methods (BIA, DXA, and 4C model) adjusted by sex and pubertal development. The 4C model showed the highest precision, while BIA had the largest bias, especially for children with less adiposity or in the earlier stages of puberty.	To minimize error, it is important to use a combination of appropriate methods to obtain reliable BC measurements. BIA method is considered less acceptable. DXA and isotopic dilution have the highest accuracy and reliability for measuring BF in obese children and adolescents.
Meredith-Jones K.A. et al., 2014 [26] New Zealand	187	No intervention	BIA (Tanita BC-418) DXA (Lunar Prodigy scanner)	The study focused on the ability of BIA to track changes in BC in young children over a 12-month period. Both methods provided similar results, with no significant differences for changes in FM or FFM.	Hand-to-foot bioimpedance accurately estimates changes in FM, FFM, and BF% over a 1-year period when compared with measurements obtained by DXA.
Luque V. et al., 2014 [27] Spain	171	No intervention	BIA (Tanita BC-418) DXA (General Electric Lunar Prodigy Advance)	Validation of BIA for estimating overall BC in 7-year-old children, comparing it to DXA as the reference method. Results showed that BIA outputs had a moderate bias for FM estimates but that BIA regressions provided more accurate and reliable predictions of FM and FFM.	BIA is a valid suport technique in clinical diagnosis and monitoring of children with overweight and obesity. The validation of raw impedance mesurements in specific populations may increase the accuracy of the technique.
Luque V. et al., 2014 [28] Spain	171	No intervention	BIA (Tanita BC-418) DXA (General Electric Lunar Prodigy Advance)	Validation of segmental BC using BIA compared to DXA, including measurements of FM and FFM in the trunk, left arm, and left leg. BIA regressions provided more accurate and reliable estimates, especially for trunk and arm measurements.	Segmental BC measurements predicted by Tanita BC-418 are not valid for clinical or epidemiological use at individual level, except for leg lean mass.
Benjaminsen C.R. et al., 2024 [29] Denmark	92	Family-centered lifestyle intervention	BIA (Tanita BC-420MA) DXA (GE Lunar iDXA 2007)	BIA effectively monitors longitudinal changes in BC at a group level, but is less reliable for individual assessments.	Suitable for group-level studies, but not individual assessments.
Dettlaff-Dunowska M. et al., 2022 [30] Poland	152	Integrated weight-loss program (dietary, psychological, and physical care)	BIA (Tanita DC-430 S MA device) DXA (Hologic)	Decrease in FM and increase in FFM. Improved physical fitness. Positive correlation between muscle mass increase and physical fitness improvement (r = 0.49 for FFM)	Both BIA and DXA methods are equally useful for measuring BC over time. BIA is more practical for routine use in clinical settings due to its ease of use and lower cost, while DXA is more accurate, but requires specialized equipment.
Zembura M. et al., 2023 [31] Poland	95	No intervention	BIA (Tanita BC480MA) DXA (Hologic) Dynamometer Six-minute walk test Timed up-and-go test	Sarcopenia prevalence (6.32% to 97.89%). The lack of standardized pediatric-specific sarcopenic obesity diagnostic criteria limits comparability and consistency of results.	BIA is affordable and portable, but relies on hydration status and indirect estimation of muscle mass through conversion equations and must be calibrated with DXA data.
Samouda H, Langlet J, 2022 [32] Luxembourg	197	No intervention	BIA device (Tanita BC-532) DXA (Hologic^®^ QDR4500W)	The study aimed to compare BF% between BIA and DXA. Results show BIA significantly underestimates FM compared to DXA, with a high degree of error.	The BIA Tanita BC-532 device is considered effective, easy to use, and portable, making it practical for screening large populations. However, due to significant underestimation of FM, BIA is not reliable enough for precise clinical diagnosis and should not replace DXA for accurate BC assessments.
Kabiri L.S. et al., 2015 [9] China	55	No intervention	BIA (Tanita BF-689) DXA (Discovery QDR-4500 for Windows; Hologic)	Primary outcome was to assess the reliability, validity, and diagnostic value of BIA compared to DXA for %BF in elementary school children.	Compared to a DXA machine, the BF-689 is affordable and portable, making it an efficient tool for assessing %BF in elementary-school-aged children.
Butcher A. et al., 2018 [33] Texas, SUA	112	No intervention	BIA (Tanita BF-689) DXA (Horizon)	The Tanita BF-689 showed from poor to good agreement with DEXA for %BF measurements, from poor to moderate agreement for tracking changes in %BF over time, high sensitivity for identifying individuals in the healthy category, and high specificity for classifying individuals as underfat, overfat, or obese.	BIA showes high specificity in classifying adolescents as obese or overfat, making it valuable for screening. However, it underestimates FM, especially in leaner adolescents, and has limited sensitivity for tracking changes in BF. This makes BIA useful for large-scale screening and health monitoring, but less reliable for assessments in populations with low or moderate levels of fat mass.
Thivel D. et al., 2018 [34] France	196	Multidisciplinary weight loss program	BIA (BIA-Tanita MC-780 DXA (Hologic) before and after a 3-month weight loss program.	Comparison of the ability of BIA and DXA to track BC changes in obese adolescents after a 3-month weight loss program. BIA is more effective for tracking FM changes in less obese individuals, while its accuracy for FFM changes was poor, especially in adolescents with severe obesity.	BIA’s precision in assessing BC declines as obesity levels rise. Its ability to consistently track changes is compromised by high initial body weight or fluctuations in weight, FM, FFM, and BMI. Results show a limitation of BIA at an individual level and that it cannot be used interchangeably with other methods such as DXA.
Verney J. et al., 2016 [35] France	138	No intervention	BIA (Tanita MC-780) multifrequency analyzer DXA (Hologic)	The study focused on comparing BIA and DXA for assessing whole-body FM and FFM. It showed that BIA overestimated FM and underestimated FFM, but was relatively accurate for obese adolescents. It highlighted the loss of correlation between BIA and DXA as adiposity increased.	Tanita MC-780 is a valuable method to determine whole-body measurements of BC. Both methods have a high level of agreement and concordance. The results can be modified in severe obesity adolescents.

**Table 2 diagnostics-15-01505-t002:** BIA using InBody devices.

Author, Year, Country	N	Intervention Type	Assessment Techniques (Devices)	Primary Outcome(s)	Clinical Utility
Khan S. et al., 2020 [36] USA	78	No intervention	BIA device (InBody 370, a stationary multifrequency octopolar) BIA (Omron handheld single-frequency tetrapolar–SF4) DXA (Hologic^®^ Horizon)	The results showed that InBody 370 MF8 BIA device is more accurate for estimating BF%, especially in severely obese children. Also, it was accurate in estimating appendicular lean mass.	The MF8 BIA device is particularly precise in estimating BF% and appendicular lean mass compared to DXA. Its point-of-care feature makes it very useful in clinics for evaluating BC in children with severe obesity.
Huang Y. et. al., 2023 [37] China	172	No intervention	BIA (InBody 720 octapolar multi-frequency) DXA (Hologic Discovery fan-beam densitometers) Air displacement plethysmography (ADP) (BOD POD system, Cosmed Inc)	BIA underestimates FM and overestimates FM% compared to DXA. The smallest bias occurs in children with obesity. Agreement decreases as BMI decreases. Regional analysis aligns with DXA for appendicular skeletal muscle mass.	BIA is cost-effective, portable, and practical for large-scale epidemiological studies. However, clinical use is limited by wider variability at the individual level. DXA remains the gold standard for accuracy in clinical and research settings.
Seo Y.G. et al., 2016 [38] Korea	316	No intervention	BIA (InBody 720 BC Analyzer) DXA scanning (Lunar Prodigy Advance)	The study found better agreement between BIA and DXA in children with severe obesity compared to those with mild/moderate obesity. The bias decreased as obesity severity increased, highlighting a more reliable use of BIA in higher BMI children.	BC analysis using BIA could be valuable for assessing the impact of interventions on children and adolescents with severe obesity in clinical settings.
Tompuri T.T. et al., 2019 [39] Finland	350	No intervention	BIA (InBody 720) DXA (Lunar Prodigy Advance)	The study shows that BIA is a useful tool for assessing adiposity and cardiometabolic risk in prepubertal children, but DXA provides more accurate results, especially for girls.	Adiposity measurements can be used as screening tools for elevated cardiometabolic risk. However, BF% assessed by BIA or DXA does not offer any advantage over traditional anthropometric measures for detecting cardiometabolic risk in prepubertal children.
Howe C.A. et al., 2021 [5] Ohio, SUA	58	No intervention	BIA (InBody 770 BIA) Resting metabolic rate (MedGem) DXA (Hologic)	Comparing BIA-derived whole-body measurements of BC to DXA, no differences were observed in BF%, fat mass index (FMI), fat-free mass index (FFMI), and visceral adipose tissue (VAT). However, on individual level, BIA showed significant differences in BF%, FMI, and FFMI among youth of a healthy weight and FMI in teenagers. Mean difference between InBody and DXA was 7.8%.	InBody 770 is a newer method that estimates total body water and is not influenced by the intake of nutrients. However, the preliminary findings indicate that when using BIA, it is important to evaluate aspects of the young person’s health and weight status with caution, particularly among boys and teenagers.

**Table 3 diagnostics-15-01505-t003:** BIA using SECA devices.

Author, Year, Country	N	Intervention Type	Assessment Techniques (Devices)	Primary Outcome(s)	Clinical Utility
Lopez-Gonzalez D. et al., 2022 [40] Mexico	450	No intervention	BIA methods: (1) standing-position BIA handrail (SECA mBCA 514), (2) standing-position BIA handle (SECA modified mBCA 514), (3) supine-position BIA (SECA 525). DXA (Lunar-iDXA densitometer)	The study validated BIA methods against DXA for assessing total body and regional BC (FM and FFM). BIA showed strong correlation, but also significant biases, particularly for FM.	All BIA methods have good levels of correlation and concordance with DXA BC estimations, but the BIA handrail has the lowest concordance.
González-Ruíz et al., 2018 [41] Colombia	127	No intervention	BIA (Seca mBCA 514, Tanita BC 420 MA) DXA (Hologic Horizon) Slaughter skinfold thickness equations	The study assessed the validity of BIA, Slaughter skinfold thickness equations, and DXA for estimating %BF in Latin American children with excess adiposity. BIA methods and Slaughter equations provided significant underestimations of BF%, with poor agreement with DXA.	BIA devices and Slaughter skinfold thickness equations, although widely used for field screening, showed limitations for accurate %BF measurement in children with excess adiposity. These methods may be useful for initial screening in large populations but should not replace DXA for precise body fat assessments, especially for clinical or research applications in which accuracy is critical.

**Table 4 diagnostics-15-01505-t004:** BIA using Bodystat Quadscan 4000 device.

Author, Year, Country	N	Intervention Type	Assessment Techniques (Devices)	Primary Outcome(s)	Clinical Utility
Noradilah M.J. et al., 2016 [42] Malaysia	160	No intervention	BIA (Bodystat Quadscan 4000) SKF DXA (Hologic QDR series)	All equations significantly underestimated %BF (*p* < 0.05). Despite BIA’s tendency to underestimate BF% compared to DXA, it proved more suitable for measuring BF% in a population similar to the study sample than SKF. This indicates a need for new SKF equations tailored to specific populations.	BIA-based prediction equation from the manufacturer had better agreement with DXA and can be used to measure BC at population level in Malay children.
Visuthranukul C. et al., 2015 [43] Thailand	52	Low-GI diet vs. control group (low-fat)	BIA(Bodystat Quadscan 4000) DXA (Hologic QDR Discovery A)	The main outcomes were the changes in BC, measured by BIA and DXA, and changes in insulin sensitivity. The low-GI diet group showed a significant reduction in fasting insulin and HOMA-IR, indicating improved insulin sensitivity compared to the control group.	When stratified by age group, the absolute biases of FM and FFM for the two methods (BIA and DXA) showed that BIA underestimates BF%, but using the same technique would not change the main outcomes between children and adolescents with a low-glycemic index.

**Table 5 diagnostics-15-01505-t005:** BIA using Quantum device.

Author, Year, Country	N	Intervention Type	Assessment Techniques (Devices)	Primary Outcome(s)	Clinical Utility
Lyra A. et al., 2015 [44] Brazil	111	Lifestyle modification program (physical activity + diet)	BIA (BIA Quantum) DXA (Lunar DPX-IQ, version 4.7e)	Comparison of FM and FFM changes before and after a lifestyle modification program. DXA detected changes in both FM and FFM, while BIA detected only FM reduction.	BIA is not effective for assessing the impact of short-term physical activity in obese adolescents. It overestimates FFM and underestimates FM compared to DXA.
Ejlerskov K.T. et al., 2014 [45] Denmark	99	No intervention	BIA (tetrapolar bioelectrical impedance Analyser Quantum III) DXA (Lunar Prodigy Advance)	The study aimed to develop and validate predictive equations for FFM using BIA and anthropometry in 3-year-old children, with DXA as the reference method. Both BIA regression models showed low level of bias and high predictive accuracy, providing reliable estimates of FFM and FM for this age group.	In this age group, BIA and anthropometry have practical advantages compared to DXA and other techniques as the measurements are easily obtained. It can prove useful for population studies linking early risk factors to BC and early onset of obesity. Predictive equation according to BIA method should be applied with caution in study settings, because the children differ considerably in age, height, and health status, which is likely to affect their hydration level.

ADP = air displacement plethysmography; BC = body composition; BF% = body fat percentage; BIA = bioelectrical impedance analysis; DXA = dual-energy X-ray absorptiometry; FFM = fat-free mass; FFMI = fat-free mass index; FM = fat mass; FMI = fat mass index; MF8 = InBody 370, a stationary multifrequency octopolar device; RMR = resting metabolic rate; SF4 = Omron handheld single-frequency tetrapolar device; SKF = skinfold; VAT = visceral adipose tissue; Slg-Eq = the Slaughter equation.

**Table 6 diagnostics-15-01505-t006:** Study design and the statistical significance analysis of included articles.

Study Design	Author, Year	N	Mean Age (Years)	Mean BMI (kg/m^2)^	*p*-Value	Correlation (r)
Randomized controlled trial	Visuthranukul C. et al., 2015 [43]	52	12.0 ± 2	33.1 ± 6.6 (control), 34.2 ± 5.8 (intervention)	*p* = 0.004 (fasting plasma insulin), *p* = 0.007 (HOMA-IR)	BIA %Fat vs. DXA %Fat: 0.77, FMI: 0.91
	Kasvis P. et al., 2014 [21]	89	9.7 ± 1.7 (girls), 10.0 ± 1.7 (boys)	Not reported (mean BMI z score: 2.86 ± 0.74)	*p* = 1.000 (FM and %BF change agreement between BIA and DXA), *p* < 0.003 (Android vs. Gynoid %BF difference), *p* < 0.013 (Android vs. Gynoid FM difference)	%BF: 0.803–0.848, FM: 0.950–0.967, FFM: 0.906–0.944 (all *p* < 0.0001)
	Lyra A. et al., 2015 [44]	111	12.0 ± 1.9	Not reported (mean BMI z-score: 2.3 ± 0.5)	*p* < 0.001 (Mann–Whitney for FM%), *p* < 0.001 (Student t-test for FFM)	BMI z-score vs. DXA FM%: 0.58 (*p* < 0.01); BMI z-score vs. BIA FM%: 0.42 (*p* < 0.01); Trunk fat DXA vs. WC/height: 0.65 (*p* < 0.01)
	Wan C.S. et al., 2014 [19]	66	12.9 ± 2	34.5 ± 5.5 (boys), 33.4 ± 5.8 (girls)	<0.001	FFM: 0.92, FM: 0.93, %BF: 0.78; Change in %BF: BIA vs. DXA: r = 0.69 (manufacturer equation), r = 0.78 (derived equation)
	de Silva M.H.A.D. et al., 2021 [20]	97	10.6 ± 2.5	25.5 ± 3.7	FM 0.001, FFM 0.018	FM: 0.92, FFM: 0.83 (*p* < 0.001 for both)
	Kabiri L.S. et al., 2015 [9]	55	8.47 ± 1.65	17.8 ± 3.4	<0.001	BIA vs. DXA ICC: 0.788 (−0.167, 0.942); Pearson’s correlation: r = 0.901 (*p* = 0.01)
Clinical trial (randomized, double-blind, placebo controlled	Dettlaff-Dunowska M. et al., 2022 [30]	152	10.93 ± 2.97	24.78 ± 3.88	<0.05	FM reduction vs. fitness improvement: −0.542, MM increase vs. fitness improvement: 0.488
Longitudinal validation study	Meredith-Jones K.A. et al., 2014 [26]	187	6.5 ± 1.5 (girls), 6.3 ± 1.4 (boys)	18.2 ± 4.4	*p* < 0.001 (FFM and FM), *p* = 0.042 (%BF in normal-weight girls)	Baseline %BF: r = 0.916, FFM: r = 0.956, FM: r = 0.974 (all *p* < 0.001); Change over 1 year: FFM: r = 0.53 (*p* < 0.001), FM: r = 0.36 (*p* < 0.001), %BF: r = 0.06 (*p* = 0.38)
Observational cohort study	Benjaminsen C.R. et al., 2024 [29]	92	10.5 ± 2.9	Not specified (BMI z-score 3.1 ± 0.8)	<0.001	FM: 0.97, FM%: 0.83, FFM: 0.98, FFM%: 0.83
Cross-sectional validation study	Huang Y. et. al., 2023 [37]	172	9.7 ± 3.1	Not reported (mean BMI z-score: boys 0.9 ± 1.7, girls 0.6 ± 1.7)	<0.001	FM: 0.964 (Boys), 0.868 (Girls); FFM: 0.976 (Boys), 0.895 (Girls)
	Lopez-Gonzalez D. et al., 2022 [40]	450	12 ± 3.7	22.4 ± 5.1	<0.001	FM: 0.99 (Handle), 0.99 (Handrail), 0.99 (Supine 8e), 0.99 (Supine 4e); FFM: 0.99 (Handle), 0.99 (Handrail), 1.00 (Supine 8e), 0.99 (Supine 4e)
	Howe C.A. et al., 2021 [5]	58	11.4 ± 2.9	16.4 ± 1.1 (healthy weight), 22.7 ± 2.9 (overweight)	<0.001	BF%: 0.96, RMR: 0.79; PA vs. BF%: −0.33 (*p* = 0.01), PA vs. FFM: 0.59 (*p* < 0.001), PA vs. grip strength: 0.56 (*p* < 0.001)
	Gutierrez-Marín D. et al., 2021 [22]	315	10.8 ± 1.6	26.0 ± 2.8	<0.001	FFMTANITA vs. FFM4C: 0.969, FFMZ vs. FFM4C: 0.968; Bias in FM estimation was reduced from 18.4% to 6.4% (*p* < 0.001)
	Martín-Matillas M. et al., 2020 [23]	92	10.0 ± 1.2	26.8 ± 3.5	<0.001	FM: 0.89–0.97, FMI: 0.86–0.97
	Butcher A. et al., 2018 [33]	112	14 ± 1.64	21.4 ± 3.35	<0.001	ICC for absolute agreement: 0.78 (0.48–0.88); ICC for absolute agreement over time: 0.71 (0.242–0.866)
	Seo Y.G. et al., 2016 [38]	316	11.5 ± 2.1	25.0 ± 5.5	<0.05	Group 1 (mild to moderate obesity): %BF: 0.774, FM: 0.970, FFM: 0.977; Group 2 (severe obesity): %BF: 0.825, FM: 0.967, FFM: 0.982
	Thivel D. et al., 2018 [34]	196	14.0 ± 0.9	35.0 ± 4.9	FM%: <0.001, FMkg: <0.001, FFMkg: 0.721	FM%: 0.41, FMkg: 0.64, FFMkg: 0.03
	Noradilah M.J. et al., 2016 [42]	160	9.4 ± 1.1	17.4 ± 4.1	<0.05	BIA Manufacturer: 0.88, BIA Houtkooper: 0.82, BIA Kushner: 0.83, BIA Rush: 0.86
	Luque V. et al., 2014 [27]	171	7 ± 1	16.47 ± 1.56	<0.001	FM: 0.943, FFM: 0.882
	Luque V. et al., 2014 [28]	171	7 ± 1	16.47 ± 1.56	<0.001	Trunk FM: 0.839, Trunk FFM: 0.141, Left arm FM: 0.775, Left arm FFM: 0.501, Left leg FM: 0.875, Left leg FFM: 0.777
	Ejlerskov K.T. et al., 2014 [45]	99	3 ± 1	15.8 ± 1.2	Full model: *p* = 0.026, Simple model: *p* = 0.004	Full model: 0.85 (FFM), Simple model: 0.84 (FFM)
Cross-sectional study	Zembura M. et al., 2023 [31]	95	12.7 ± 3	Not reported (BMI z-score 2.91 median)	<0.05	SMMa: 0.89, FM: 0.91
	Samouda H, Langlet J, 2022 [32]	197	11.8 ± 2.3 (boys), 12.1 ±2.4 (girls)	28.2 ± 4.9 (boys), 28.3 ± 5.6 (girls)	<0.0001	Boys: 0.617, Girls: 0.648
	Khan S. et al., 2020 [36]	78	14.8 ± 2.7	36.7 ± 7.5	<0.0001 (SF4 vs. DXA), 0.001 (MF8 vs. DXA)	SF4 vs. DXA: BF%: 0.82, FM: 0.97, ICC: 0.39; MF8 vs. DXA: BF%: 0.90, FM: 0.99, ICC: 0.87
	González-Ruíz et al., 2018 [41]	127	12.9 ± 1.2 (boys), 13.7 ± 1.7 (girls)	24.2 ± 2.5 (boys), 23.5 ± 4.1 (girls)	<0.001	Boys: DXA vs. Seca^®^ mBCA 514: 0.726, DXA vs. Tanita^®^ BC 420MA: 0.430, DXA vs. Slaughter: 0.532; Girls: DXA vs. Seca^®^ mBCA 514: 0.846, DXA vs. Tanita^®^ BC 420MA: 0.652, DXA vs. Slaughter: 0.711
	Vásquez F. et al., 2016 [24]	61	8–13	Not specified	<0.05	Boys (Tanner I and II): 0.352; Boys (Tanner III and V): 0.721; Girls (Tanner I and II): 0.516; Girls (Tanner III and V): 0.754
	Verney J. et al., 2016 [35]	138	14 ± 1.5	33 ± 4.8	<0.001	FM%: 0.779, FM (kg): 0.933, Trunk FM%: 0.718, FFM (kg): 0.847, Trunk
	Tompuri T.T. et al., 2019 [39]	350	8.9 ± 1.5	17.8 ± 3.4	<0.001	Girls: BIA BF%: 0.801, DXA BF%: 0.763; Boys: BIA BF%: 0.828, DXA BF%: 0.839

## Data Availability

The data used in the study will be available from the corresponding authors upon request. The review does not have a protocol prepared.

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
