# Peer review of "Bioelectrical Impedance Analysis Versus Dual X-Ray Absorptiometry for Obesity Assessment in Pediatric Populations: A Systematic Review"

_diagnostics, 2025, doi:10.3390/diagnostics15121505_

Round 1
Reviewer 1 Report
Comments and Suggestions for Authors
The manuscript requires corrections and additions in the following areas:
Introduction - please describe the principles of analyzers/methods in more detail. Indicate the differences in devices available on the market (e.g. MF-BIA, SF-BIA). Explain which parameters are measured and which are estimated, for which populations they are intended. DXA is not the gold standard for BC, but for bone density measurement;
Methods - add data on equipment manufacturers, equations used to measure body composition for each of the devices;
Results - the attached Table 1 is illegible, requires modification and should be preceded by a description. The collected articles should be segregete by type of device. In turn, in Table 2 they should be arranged depending on the importance of the study design.
As a consequence, the discussion should be adapted to the appropriately analyzed results and refer to specific analyzers.
The conclusions are too general - do not treat all BIA devices together. Maybe it is worth limiting ourselves to specific ones.
Author Response
Dear Reviewer,
Thank you for your time and feedback.
Comments and Suggestions for Authors
The manuscript requires corrections and additions in the following areas:
Introduction - please describe the principles of analyzers/methods in more detail. Indicate the differences in devices available on the market (e.g. MF-BIA, SF-BIA).
R:Thank you for your valuable and thoughtful feedback. In response, we have introduced the classification of Bioelectrical Impedance analysis according to the frequency in lines 78-81.
Explain which parameters are measured and which are estimated, for which populations they are intended.
R: Also, we specified and explained in lines 87-94.
DXA is not the gold standard for BC, but for bone density measurement;
R:We value your feedback and reformulated the definition in rows 60-63.
Methods - add data on equipment manufacturers, equations used to measure body composition for each of the devices;
R:To improve clarity and enhance scientific precision, we added the data in lines 147-155.
Results - the attached Table 1 is illegible, requires modification and should be preceded by a description.
R: Added the description in lines 182-189.
The collected articles should be segregete by type of device. In turn, in Table 2 they should be arranged depending on the importance of the study design.As a consequence, the discussion should be adapted to the appropriately analyzed results and refer to specific analyzers.
R:Thank you for your helpful comment. In response, we have reorganized the original single table into five separate tables (Tables 1–5), each categorized by the specific BIA device manufacturer (Tanita, InBody, SECA, Bodystat, and Quantum). This structure allows for a clearer comparison of study characteristics and findings based on the device used. Each table now presents detailed information on country, sample size, intervention type (if applicable), assessment techniques (BIA and DXA), primary outcomes, and clinical utility. This approach enhances clarity and highlights the methodological heterogeneity and device-specific differences in accuracy and applicability, which are critical for interpreting agreement between BIA and DXA in pediatric populations. We also rearranged Table 2 according to the importance of the study design, following a generally accepted hierarchy of evidence: Randomized Controlled Trials (RCTs), Clinical Trials (including randomized, double-blind, placebo-controlled), Longitudinal Studies, Observational Cohort Studies, Cross-Sectional Validation Studies, and Cross-Sectional Studies.
The conclusions are too general - do not treat all BIA devices together. Maybe it is worth limiting ourselves to specific ones
R:We appreciate your thoughtful feedback and have incorporated your suggestions and reformulated conclusions into the revised manuscript.
Reviewer 2 Report
Comments and Suggestions for Authors
The topic is current and relevant, but the manuscript needs some adjustments to be accepted for publication.
Abstract
Some aspects should be reviewed:
Present the methods of the Systematic Review more clearly, describing inclusion and exclusion criteria.
It is important to correct the statement that DXA is the gold standard, replacing it with a reference method, since it is a three-compartment method (3C) and only four-compartment methods (4C) can be considered the gold standard (doi: 10.1111/obr.12261).
The conclusion presented does not show the conclusions of the study; it is only reporting the objectives.
Introduction
Review the statement that DXA is the gold standard. In addition, the issue of radiation exposure cannot be considered a limitation of the method, since it is minimal.
In the objective (last paragraph), I suggest replacing doctors with health professionals, since the assessment of body composition is also widely used by nutritionists and physical education professionals.
Methods
If the goal was to include children and adolescents, why limit the maximum age to 17 years, if adolescence goes up to 19 years? I suggest consulting doi: 10.3389/fnut.2022.820736. Justify the limitation of the maximum age.
Discussion
Most of the included studies used equipment that does not allow the identification of the predictive equations available in their software, which Campa et al. (doi: 10.1186/s12967-024-05272-x) call “black box”, such as those of the brands (Tanita, InBody, Seca and Omron). Do these equipments have equations developed and validated for children and adolescents with obesity? This limitation of the equipment should be clearly stated in the discussion.
In lines 241 to 244, a statement is made that deserves to be referenced. I suggest: doi: 10.1038/s41430-021-00946-x and doi: 10.1038/s41430-023-01310-x.
Regarding the last paragraph, do the authors really believe that BIA could become a standard method, since it is a doubly indirect technique?
At times the text seems a little truncated to me. A spelling and writing review could improve it.
Author Response
Dear Reviewer,
Thank you for your time and feedback.
Comments and Suggestions for Authors
The topic is current and relevant, but the manuscript needs some adjustments to be accepted for publication.
Abstract
Some aspects should be reviewed:
Present the methods of the Systematic Review more clearly, describing inclusion and exclusion criteria.
It is important to correct the statement that DXA is the gold standard, replacing it with a reference method, since it is a three-compartment method (3C) and only four-compartment methods (4C) can be considered the gold standard (doi: 10.1111/obr.12261).
The conclusion presented does not show the conclusions of the study; it is only reporting the objectives.
R:Thank you for your valuable and thoughtful feedback. In response, we have revised the Abstract to improve clarity and enhance scientific precision.
Introduction
Review the statement that DXA is the gold standard. In addition, the issue of radiation exposure cannot be considered a limitation of the method, since it is minimal.
R:We have reviewed and revised the definition in lines 60–63 for improved clarity. Additionally, we included our concern regarding the potential risk of irradiation in lines 66–72.
In the objective (last paragraph), I suggest replacing doctors with health professionals, since the assessment of body composition is also widely used by nutritionists and physical education professionals.
R:We value your feedback and replaced accordingly.
Methods
If the goal was to include children and adolescents, why limit the maximum age to 17 years, if adolescence goes up to 19 years? I suggest consulting doi: 10.3389/fnut.2022.820736. Justify the limitation of the maximum age.
R:Thank you for your suggestions, but many studies had the inclusion criteria up to 17 years old (e.g. https://doi.org/10.3390/ijerph181910094, DOI: 10.1186/s12902-022-01111-6, DOI: 10.1089/chi.2018.0183 and others). Although adolescence is defined by the World Health Organization as extending up to 19 years of age, this systematic review limited the age range to 2–17 years to maintain consistency with the inclusion criteria commonly used in the included studies. Additionally, many national health systems and institutional review boards define 18 years and above as adult. The exclusion of 18–19-year-olds also avoided heterogeneity introduced by potential legal, physiological, and lifestyle transitions associated with adulthood, such as changes in body composition due to late puberty completion or independent living. We acknowledge the broader definition of adolescence as proposed in literature (e.g., Front Nutr. 2022;9:820736), but for methodological clarity and comparability, we focused on the age group most consistently classified as pediatric across studies.
Discussion
Most of the included studies used equipment that does not allow the identification of the predictive equations available in their software, which Campa et al. (doi: 10.1186/s12967-024-05272-x) call “black box”, such as those of the brands (Tanita, InBody, Seca and Omron). Do these equipments have equations developed and validated for children and adolescents with obesity? This limitation of the equipment should be clearly stated in the discussion.
In lines 241 to 244, a statement is made that deserves to be referenced. I suggest: doi: 10.1038/s41430-021-00946-x and doi: 10.1038/s41430-023-01310-x.
Regarding the last paragraph, do the authors really believe that BIA could become a standard method, since it is a doubly indirect technique?
R:We appreciate your thoughtful feedback and have incorporated your suggestion into the revised manuscript (lines 303–312). In this section, we now explicitly address the limitations of BIA devices, particularly the use of proprietary, non-transparent algorithms and their implications for pediatric populations with obesity. We also cite relevant literature to support the need for device-specific validation and clarify BIA’s role as a complementary, not replacement, tool to DXA (lines 321-326).
Comments on the Quality of English Language
At times the text seems a little truncated to me. A spelling and writing review could improve it.
R: We improved English Language
Round 2
Reviewer 1 Report
Comments and Suggestions for Authors
Thank you for considering and taking into account my comments, we will not report any more